# Mental Health Literacy Intervention on Help-Seeking in Athletes: A Systematic Review

**DOI:** 10.3390/ijerph17197263

**Published:** 2020-10-04

**Authors:** Danran Bu, Pak-Kwong Chung, Chun-Qing Zhang, Jingdong Liu, Xiang Wang

**Affiliations:** 1Department of Sport, Physical Education and Health, Hong Kong Baptist University, Hong Kong, China; 17481791@life.hkbu.edu.hk (D.B.); cqzhang@hkbu.edu.hk (C.-Q.Z.); 2HuBei Institute of Sport Science, Wuhan 432025, China; 3Department of Psychology, Sun Yat-Sen University, Guangzhou 510275, China; 4Department of Physical Education, Sun Yat-Sen University, Guangzhou 510275, China; liujd7@mail.sysu.edu.cn; 5Graduate School, Wuhan Sports University, Wuhan 430079, China; wangxiang199207@outlook.com

**Keywords:** mental health literacy, help-seeking, athletes, systematic review

## Abstract

Mental health literacy (MHL) is recognised as a major factor in whether athletes seek help when they experience mental health difficulties. Therefore, the current study aimed to provide a systematic review of the effectiveness of MHL training programmes in improving mental health knowledge and help-seeking and reducing stigma among athletes. To identify intervention studies of MHL programmes, five electronic databases were systematically searched for articles published before May 2020. The selection procedure was based on the Preferred Reporting Items for Systematic Reviews and Meta-Analyses (PRISMA) guidelines. All kinds of study designs were included. Effect sizes were calculated for mental health knowledge, stigma reduction and help-seeking attitudes, intentions and behaviours. Risk of bias was assessed for each study using the Cochrane tool and the Newcastle–Ottawa quality assessment scale. Five studies (1239 participants in total) were selected for review. Overall, either small or medium effects were found for mental health knowledge, stigma reduction, help-seeking attitudes, and intentions for post- and follow-up interventions, whereas a null effect was found in help-seeking behaviours for both post- and follow-up interventions. Furthermore, three studies had a low risk of bias, and two had a high risk of bias. MHL interventions can enhance help-seeking attitudes and intentions and mental health knowledge and reduce stigma but do not increase help-seeking behaviours for now. Further studies should evaluate interventions to enhance help-seeking behaviours. Furthermore, the methodological quality of studies, including randomized controlled trials and other designs, should be improved in future research.

## 1. Introduction

The International Olympic Committee’s (IOC) consensus statement has affirmed the importance of mental health for elite athletes [1]. There is much evidence that the athletic population suffers from mental health issues such as depression and anxiety, which affect their performance and personal development [2]. For instance, Gulliver et al. [3] found that 23.2% of the Australian elite athletes surveyed suffered from depression, while 7.1% met the criteria for generalised anxiety disorder and 4.5% for panic disorder. In another study, Gouttebarge et al. [4] revealed that 45% of the Dutch elite athletes surveyed and 29% of former elite athletes suffered from anxiety and depression. Yet, even when their suffering is severe, it seems that athletes rarely seek professional services when they experience mental health issues [5,6]. For example, Guo [7] reported that 86% of the surveyed Chinese elite athletes did not seek professional assistance when experiencing mental health issues.

Mental health help-seeking is an adaptive coping process that is the attempt to obtain assistance to treat mental health issues [8]. The model of help-seeking [9] reveals that it is a dynamic four-step process: becoming aware of a problem, expressing the problem to others, identifying appropriate and accessible help sources, and seeking professional help. In this framework process, knowledge and recognition of mental health disorders, known as mental health literacy (MHL), play a critical role. Moreover, the help-seeking model focuses on three processes: help-seeking attitudes, intentions, and behaviours [10]. A previous systematic review demonstrated that a low level of MHL is the main barrier to seeking assistance [11]. MHL is defined as knowledge and beliefs about mental disorders, which facilitate their recognition, management or prevention [12]. A low level of MHL has been demonstrated in many countries, such as Australia, Canada, and the United States [13,14]. A similar situation has been found in elite athletes [15,16]. Recent evidence has shown that enhanced MHL may contribute to help-seeking [15,17]. Therefore, MHL has received global attention as a precondition for mental health intervention [18].

In addition to the issue of low levels of MHL, high levels of stigma are recognised as another barrier influencing help-seeking in elite sport [11]. The term ‘stigma’ refers to negative thoughts, feelings, and behaviours toward individuals or groups that possess characteristics or engage in behaviours that are viewed as unacceptable by wider society [19,20]. For athletes, the most relevant stigma is being regarded as weak by coaches, teammates, parents, competitors, and even themselves, if they are identified as needing or seeking help to deal with mental problems [11,21]. These considerations might prevent them from seeking assistance. Therefore, reducing stigma is another target of some intervention programmes that aim to enhance help-seeking [22].

Due to the IOC’s increasing awareness of the importance of addressing mental health issues [1], several programmes have been developed to address these barriers and promote help-seeking. As such, a systematic review of the existing evidence is required to evaluate the efficacy of MHL interventions. Recently, a systematic review of interventions focused on athletes, coaches and officials was conducted to explore awareness of mental health and well-being [23]. Of these studies, six reported an enhancement of mental health knowledge; three reported an increase of mental health referral efficacy; and another three studies demonstrated an increase in well-being. However, no review has focused on athletes’ MHL programmes to enhance help-seeking.

In response to increasing MHL interventions in the sport context and to improve the methodological transparency of previous MHL intervention studies [24], we conducted a systematic review of MHL programmes that focused on athletes and aimed to enhance mental health knowledge, improve help-seeking attitudes, intentions and behaviours and reduce stigma. In this systematic review, we assessed MHL intervention studies with a pre-test and post-test design, including RCTs and one group intervention. We also examined the quality of the interventions reported in these studies. We did not choose to conduct a further meta-analysis, due to the diversity of the interventions and outcome measures.

## 2. Methods

### 2.1. Search Strategy

This systematic review protocol was registered on PROSPERO (registration number: CRD42020171090). The systematic study selection procedure was based on a ‘Preferred Report in Items for Systematic Reviews and Meta-analyses’ PRISMA guideline [25] (see Appendix A). We conducted search at five databases: PsycINFO, PubMed, Web of science, Scopus and SPORTDiscus. Articles published before 8 May 2020 were included, which is the updated search date. Combined keyword searching were used: ‘mental health literacy OR mental health awareness OR mental health knowledge OR mental health first aid OR mental health service OR mental health strategy OR mental health promotion OR mental health education OR mental health prevention’ AND ‘sport organisation OR sport club OR sport school OR sport setting OR sport training’ AND ‘athlete OR player OR sportsman’ AND ‘intervention OR training OR trial OR programme’ (see Appendix A).

### 2.2. Eligibility Criteria

In the current review, we focused on articles published in English. Specific inclusion and exclusion criteria were set to screen the articles for eligibility by reading the title, abstract and full text. Our inclusion criteria were (1) intervention studies, with various designs: between-subjects design (i.e., posttest-only control group design), within-subjects design (i.e., pretest-posttest design) and mixed design (i.e., pretest-posttest control group design) [26]. (2) In terms of participants, we only focused on athletes in a professional or amateur sport club or organisation. The study participants were of all ages. Our exclusion criteria were participants recruited from participating community sporting clubs, but they are not athletes [23].

### 2.3. Study Selection

The data selection was conducted as follows. First, all database searches were exported from the software to a folder. All titles, abstracts, and keywords were screened to exclude irrelevant studies. All relevant studies were retained for further screening. Then, the full text of the remaining relevant studies was assessed using the inclusion criteria. This study selection was evaluated by two independent reviewers. Discrepancies were discussed by the two reviewers until agreement was reached.

### 2.4. Data Collection Process and Data Items

The study characteristics and outcomes of each selected study were extracted. The information extracted from each selected study included the following: (1) the participants’ demographic information (including age (Mean (M), standard deviation (SD), range), percentage of women, type of sport and country); (2) intervention information (including duration, intervention deliverer, number of sessions, frequency, and delivery mode [face to face, online, blended]); and (3) study information (including design (RCT, controlled, cluster-RCT), comparison condition (waiting list group, no intervention, other educational interventions, other mental health educational interventions), sample type (athletes, student-athletes), and sample size (total, intervention/control, baseline/post/follow-up)) [27].

### 2.5. Risk of Bias Assessment

The quality of the studies was assessed according to the Cochrane Collaboration to evaluate methodological quality in systematic reviews [26]. The included studies were classified as randomised or non-randomised designs, and the design of each study was assessed with an applicable bias assessment tool. The Cochrane Collaboration tool was used to evaluate risk of bias for RCTs [26].

These criteria involved seven domains of bias: random sequence generation, allocation concealment, blinding of participants and personnel, blinding of outcome assessment, incomplete outcome data, selective reporting, and other bias. In each domain, the overall risk of bias was rated as high, low or unclear. In addition, the Newcastle–Ottawa quality assessment scale was used to assess risk of bias for non-RCTs. In each evaluation, high quality choices were identified with a star [28]. Specific, selection, comparability and exposure (for case-control studies)/outcome (for cohort studies) with four, one and three questions, respectively, were evaluated. A study could be assigned a maximum of one star for each numbered item in the selection and exposure/outcome categories. A maximum of two stars could be awarded for the comparability category. According to the recommendations of the Cochrane Collaboration, the risk of bias of each study based on the overall bias results for each outcome was reported [23]. As with the study selection, the risk of bias for all included studies was evaluated by two independent reviewers, and a third reviewer was consulted and consensus was reached in the event of disagreement.

### 2.6. Outcome Categories

Three sets of outcomes were evaluated: (1) mental health knowledge/literacy/awareness; (2) help-seeking attitudes, and intentions and behaviours (according to the spectrum of activity, help-seeking process involves attitudes, intentions and actual behaviours, which further supported the theory of planned behaviour [10]). According to their approaches to help, outcomes were further classified as formal help, informal help and self-help [8,29]; and (3) stigma or negative attitudes toward mental health problems.

### 2.7. Analysis

To assess the effects of these interventions, the names of the measures and the reported values for intervention effectiveness (i.e., *p* value, effect size) were collected and calculated. Following previous studies, the statistically significant effects were based on *p* < 0.05 and the effect size *d* was classified as small = 0.2, medium = 0.5, or large = 0.8 [30]. For help-seeking behaviour, the odds ratio (OR) was provided. Given that this is a systematic review and only a limited number of studies were included, we did not conduct any further subgroup and sensitivity analyses.

## 3. Results

A flowchart presenting the procedure for selecting the studies is shown in Figure 1. The initial database search identified 1338 studies. After removing 167 duplicates, the titles and abstracts of 1171 potentially eligible studies were screened, and 1118 were excluded. After screening the full text of the remaining 53 studies, only five studies met the inclusion criteria. Then, the reference lists of these studies were screened. No additional studies met the inclusion criteria. In the end, five studies were used for this review.

### 3.1. Characteristics of the Included Programmes and Their Studies

The study characteristics are detailed in Table 1. The publication years ranged from 2010 to 2018. In the 5 studies, 1239 participants were involved, including 450 women and 279 men. Three studies [17,31,32] did not report the gender of the participants, and involved 231, 275, and four participants, respectively. The country where the studies were carried out were Australia (*n* = 2), the US (*n* = 2), and Ireland (*n* = 1). The participants were elite athletes (*n* = 2), and student-athletes (*n* = 3). Three studies reported the type of sport [15,31,32], and two studies did not [17,22].

For each study, the names of the authors, the year of publication, the programme name, the delivery mode, the follow-up points, and the duration are summarised in Table 2. The interventions implemented were the State of Mind Ireland (SOMI) (*n* = 1) [15], the Elite Athlete Mental Health Strategy (TEAMS) (*n* = 1) [22], Athletes Connected (AC) (*n* = 1) [17], Mental Health First Aid (MHFA) (*n* = 1) [31] and Support For Sport (*n* = 1) [32]. Four of these studies were designed by the intervention group [15,17,22,32] and one was based on an existing protocol (MHFA) [31]. The study designs included RCTs (*n* = 2), non-RCTs (*n* = 2) and pre-test–post-test designs with no control group (*n* = 1). All studies used a variety of comparison conditions, with three using active programmes and one using a non-intervention design. The modes of delivery were online (*n* = 2) and face to face (*n* = 3). The Gulliver et al. study [22] carried out a follow-up test three months after the initial intervention (*n* = 1); the other interventions did not [15,17,31,32]. The Breslin et al. study [15] did not provide data for all of the variables due to a low participation rate for the three-month follow-up.

### 3.2. Study Content of the Included Programmes and their Studies

The content of the interventions in the included studies is shown in Table 3. Mental health and risk factors were highlighted in all five studies. Information on help-seeking and stigma was introduced in five [15,17,22,31,32] and four [15,17,22,32] studies, respectively. Mindfulness practice and resilience were included in the Breslin et al. study [15], two effective approaches to depression were proposed in the Gulliver et al. study [22], and how to support friends was discussed in the Kern et al. study [17]. Information on referral was summarised in the study by VanRaalte et al. [32].

### 3.3. Study Results of Included Programs and Their Studies

The results of the included studies are summarised in Table 4 and Table 5. MHL, help-seeking, and stigma-related outcomes were evaluated by four, four and three studies, respectively. Other outcomes were also examined, including well-being (*n* = 1) [15], resilience (*n* = 1) [15], psychological distress (*n* = 1, reported in the pre-test but not in the post-test and follow-up test) [22], supporting teammates (*n* = 1) [17], and self-efficacy (*n* = 1) [32]. The effects of the programmes on each key outcome are discussed below.

#### 3.3.1. Effects on Mental Health Knowledge

Four eligible studies [15,17,22,32] described the results on mental health knowledge (see Table 4). Five instruments were used to evaluate knowledge. Among these four studies, two studies [17,32] reported statistically significant improvements in the level of MHL in athletes. The Gulliver et al. study [22] reported significant improvements in depression literacy and anxiety literacy in the post-test and the three-month follow-up test. The Breslin et al. study [15] evaluated different dimensions of MHL and reported significant improvements in four dimensions, but not in the other dimensions. Three studies [15,22,32] reported small to large effect sizes. Finally, two studies [15,17] were based on dichotomous outcome measures and did not report effect sizes.

#### 3.3.2. Effects on Help-Seeking

As shown in Table 4, four studies [15,17,22,31] included help-seeking outcomes. None of the measures to evaluate help-seeking attitudes, intentions and behaviour were the same across the studies. Of these four studies, three evaluated help-seeking attitudes [17,22,31], two studies [15,22] examined help-seeking intentions, and one study [22] tested help-seeking behaviour, including formal and informal approaches.

For help-seeking attitudes, three studies reported improvements (see Table 4). The Gulliver et al. study [22] reported that help-seeking attitudes in the MHL/destigmatisation condition were not significant compared with the control group in the post-intervention or the three-month follow-up, with small effect sizes. The Pierce et al. study [31] showed significant improvements in help-seeking attitudes from different sources but did not provide an effect size. The Kern et al. Study [17] reported that the participants significantly increased their help-seeking intentions, with a medium effect size.

In terms of help-seeking intentions (see Table 4), The Breslin et al. study [15] reported that the participants significantly increased their help-seeking intentions, and the Gulliver et al. study [22] showed that the intentions to seek help from formal and informal sources were not significant for the MHL/destigmatisation condition compared with the control group in the post-intervention or three-month follow-up. Small effect sizes were reported in the three studies mentioning help-seeking intention outcomes.

Changes in actual help-seeking behaviour in the post-test and follow-up test, including both formal and informal approaches, were only reported in the Gulliver et al. [22]. The results were not significant for the MHL/destigmatisation condition compared with the control group in the post-intervention or three-month follow-up. The OR and 95% CI are presented in Table 4.

#### 3.3.3. Effects on Stigma

Three studies [17,22,31] discussed stigma using four types of measures. Two studies [17,22] reported that the intervention had a significant positive effect on the level of stigma of the participants, and the Pierce et al. study [31] showed no significant reduction. The Gulliver et al. study [22] found that two types of stigma, anxiety stigma and depression stigma, were significantly reduced in the post-intervention and three-month follow-up tests. Both studies reported small to medium effect sizes for stigma (see Table 4).

#### 3.3.4. Effects on Additional Outcomes

In the five studies identified, additional outcomes were examined, including well-being, resilience, psychological distress, supporting teammates and self-efficacy (see Table 5). Psychological distress was only reported in the pre-test [22]. The Breslin et al. study [15] showed that well-being and resilience were not significantly enhanced, with small effect sizes. The Kern et al. study [17] reported significant improvements in supporting teammates, with small to medium effect sizes. Finally, the VanRaalte et al. study [32] reported significant improvements in self-efficacy, with a medium effect size.

### 3.4. Risk of Bias in Included Studies

Following the Cochrane guidelines, the risk of bias in the eligible studies with RCTs and non-RCTs is summarised in Table 6 and Table 7, respectively. For the two RCT studies [22,32], random sequence generation was conducted in both studies, but one did not report the randomisation method [32]. The methods of allocation concealment were mixed: the Gulliver et al. study [22] provided specific information and was rated as low risk of bias, and the Van Raalte et al. study [32] did not provide this information and was rated as unclear. As with allocation concealment, the methods for blinding participants and personnel were mixed: the Gulliver et al. study [22] with blinded participants was rated as low risk of bias, and the Van Raalte et al. study [32] was rated as unclear. However, outcome assessment was discussed in the two studies classified as unclear for blinding. Missing data, selective reporting and other bias were controlled by both studies with low risk of bias. In summary, these two studies were rated as low risk [22] and unclear risk of bias [32], respectively.

Regarding the two non-randomised studies [15,31] and the study with the pre-test–post-test design [17], according to the Newcastle–Ottawa quality assessment scale [28,33], the study by the Breslin et al. [15] with seven stars and that by the Kern et al. [17] with six stars were classified as low risk, and the Pierce et al. study [31] with three stars was classified as high risk of bias. For the two non-randomised studies, the case definition, the representativeness of the cases and the selection of the controls were discussed. The definition of the controls, comparability (the study controls for age), the same method of ascertainment for the cases and controls and the non-response rate were only satisfied by the Breslin et al. [15]. Exposure ascertainment was not achieved by either study [15,31]. For the pre-test–post-test design study [17], the representativeness of the exposed cohort (whether it is truly representative of the average gender and education in the community), the selection of the non-exposed cohort, ascertainment of exposure, and the independent blind assessment of outcomes were achieved. However, comparability (the study does not control for the most important factor), follow-up long enough for the outcomes to occur and adequacy of follow-up of cohorts were not reported.

### 3.5. Outcome Measure Validity Assessment

Most studies measured MHL objectively, usually using true or false questions, with two studies [15,22] reporting acceptable reliability and two studies [17,32] not describing reliability and validity. In addition, one study [15] evaluated different dimensions of MHL, and one dimension was measured using a Likert scale with good reliability and validity (α = 0.55).

Similarly, for the help-seeking instruments, two studies [15,22] had good reliability and validity, and the other two [17,31] did not provide these statistics. In addition, stigma was measured using three types of measures, and two studies [17,31] did not report evidence of reliability and validity. Finally, for additional outcomes, only resilience in the Breslin et al. study [15] was assessed for reliability and validity; the other outcomes were not.

## 4. Discussion

The purpose of this systematic review was to provide evidence of the effectiveness of MHL interventions enhancing mental health knowledge, and help-seeking attitudes, intentions, and behaviours, as well as reducing stigmas in athletes. All studies meeting the inclusion criteria were systematically reviewed to offer recommendations to researchers in the process of designing and evaluating MHL studies.

We searched a range of electronic databases and identified five studies: two RCTs, two non-RCTs and one pre-test–post-test design study with no control group. It was difficult to draw conclusions about MHL interventions from these studies due to the small number of studies and the considerable heterogeneity between the interventions used regarding content and delivery. In addition, the frequency and duration of the sessions for each programme varied from 10 min [32] to 12 h [31]. Moreover, only one study [22] implemented a follow-up test, making it impossible to determine whether the improvements due to the interventions were maintained over time. Furthermore, for the two RCT studies, one was classified as unclear risk of bias [32] and the other as low risk [22]. For the non-randomised studies, the risk of bias of two studies was low [15,17], while that of one study was high [31]. As a result, there was insufficient evidence to report what type of programme was most effective or promising.

### 4.1. Effects of Studies on Mental Health Knowledge Outcomes

There was positive evidence that MHL training improved mental health knowledge, such as recognition of depression or anxiety disorders and referral knowledge in four interventions [15,17,22,32]. Only two studies reported effect sizes, limiting the explanation of the interventions [22,32]. Depending on the calculations of the effect sizes, this could lead to small to large improvements in knowledge. The accurate identification of a person suffering from depression or anxiety was enhanced up to three months after the interventions, and the effects were still significant [22]. It should be noted that the Gulliver et al. study [22] was the only study with follow-up results. Thus, due to the lack of methodological rigour across the studies, it was difficult to draw conclusions about the long-term effect of sport-based MHL training on the improvement of mental health knowledge. Further research is needed to test whether MHL programmes are an effective approach to promote mental health in the athletic population.

### 4.2. Effects of the Studies on Help-Seeking Outcomes

Two studies revealed non-significant improvements in help-seeking attitudes towards mental health disorders in the short [22,31] and long terms [22]. Gulliver et al. [22] reported a small effect size for help-seeking attitudes in the post-test and a null effect in the follow-up test. In addition, Kern et al. [17] reported a medium effect size for help-seeking attitudes in the short term.

For help-seeking intentions, the Breslin et al. study [15] found an increase in intentions to help those suffering from mental health issues in the short term, with a small effect size. The Gulliver et al. study [22] also had a null effect for help-seeking intentions from informal sources in the short and long term and a small effect size for formal channels in the short and long term. Only the Gulliver et al. study [22] reported a null effect for help-seeking behaviour, both informal and formal, in the short and long terms. Overall, except for help-seeking attitudes in the long term and help-seeking intentions from informal sources in the short and long terms [22], there was no evidence of negative effects on help-seeking from the interventions.

Help-seeking behaviour was only evaluated in the Gulliver et al. study [22], which showed no improvement in informal and formal behaviours. This is consistent with previous systematic reviews related to help-seeking in another field [34]. This may be due to the fact that help-seeking behaviour is expected if people experience mental health symptoms and are therefore likely to feel the need to seek professional help [22]. As recommended by the World Health Organization (WHO) [35], informal help and self-help are necessary to meet all aspects of mental health needs. It should be noted that there was insufficient evidence of the effects of the interventions on informal help-seeking. Moreover, research has showed that some athletes prefer to deal with mental health issues through self-reliance [36,37]. Future interventions should facilitate the use of self-help strategies and informal approach programmes or materials [34,37,38].

### 4.3. Effects of the Studies on Stigma Outcomes

Three studies discussed stigma [17,22,31]. The results of two studies demonstrated a reduction in the stigma in the short [17,22] and long term [22], with small to medium effect sizes. The third study [31] revealed no significant reduction in stigma. In addition, the validity of only two measures was acceptable in one study [22] with low risk of bias, and the other two [17,31] did not provide evidence of instrument validity (high risk of bias).

Previous results have revealed that stigma, including public stigma and self-stigma, are the main barriers to seeking professional help [39,40,41]. Unfortunately, the interventions reporting effects on stigma did not identify the types of stigma. An important step for future MHL research in sport will be to evaluate whether a higher level of MHL in athletes can reduce public stigma and self-stigma to further enhance the adoption of appropriate help-seeking behaviour. In addition, with regard to methodological problems related to measurement, the results of this review suggested that training athlete role models to overcome stereotypes and provide destigmatising materials in programmes is an acceptable intervention method to reduce stigma [15].

### 4.4. Effects of the Studies on Additional Outcomes

Additional outcomes, including well-being, resilience, psychological distress, supporting teammates and self-efficacy, were examined in the five eligible studies. However, psychological distress was only reported in the pre-test and not in the post-test and follow-up test [22]. In terms of positive mental health outcomes, well-being and resilience did not significantly improve after the intervention in the Breslin et al. study [15]. The assumption that enhancing MHL can contribute to improving mental health outcomes was not supported in this review. This is consistent with the results of a systematic review in the school setting [42]. Future studies should test the robustness of research designs based on the Cochrane approach [26] to verify the effects of mental health outcomes in MHL interventions.

In terms of confidence in helping others or seeking help, two studies provided evidence for two aspects. First, for supporting teammates, the Kern et al. study [17] demonstrated that the intervention was beneficial in enhancing the confidence of the participants in their ability to support their teammates, with small to medium effect sizes. Second, regarding self-efficacy, the Van Raalte et al. study [32] found that the intervention was conducive to improving confidence in providing and seeking help, with a medium effect size. In line with research with students, confidence in helping other students with mental health issues has been reported as one of the outcomes of several studies [43,44,45,46,47]. However, the relationship between confidence in supporting others and actual helping behaviour requires further exploration [43]. One longitudinal study by Rossetto et al. [48] showed that, compared with individuals with low confidence in helping, those with high confidence in helping people with mental health issues can predict their subsequent behaviour. However, research on how confidence turns into actual helping behaviour is still needed [43]. In addition, the measure for supporting teammates was developed by the Kern et al. study team [17] and had a high risk of bias. Therefore, further studies are needed to validate the questionnaires.

### 4.5. Improving the Methodological Quality of Studies

Although the number of studies on MHL programmes for athletes has increased, only two randomised studies have been conducted. A randomised study design is believed to provide relatively reliable evidence [43]. However, in this study, one RCT study [22] was classified as low risk of bias and another [32] was classified as unclear (see Table 6). In terms of blinding the outcome assessment, the methodological concerns related to this unclear risk of bias should be addressed. The mixed results in terms of random sequence generation, allocation concealment and blinding of participants and personnel should also be addressed. Future RCT designs should follow the CONsolidated Standards of Reporting Trials (CONSORT) procedure [49] to reduce the risk of bias based on random sequence generation, allocation concealment, blinding of participants and personnel and assessment of outcomes.

Two studies with a controlled trial and one with a pre-test–post-test design were evaluated using the Newcastle–Ottawa quality assessment scale [33]. Improving the quality of studies with non-randomised designs is crucial to create an evidence base for MHL programmes [43]. According to the Newcastle–Ottawa quality assessment [33], confounding variables should be controlled. In addition, participants, outcome assessors, and withdrawal rates should be blinded. For possible confounding variables, previous studies have shown that gender can be associated with a low level of MHL, a high level of stigma and a lower likelihood of help-seeking [50,51], which should be considered when controlling for confounding variables. Several methods can be used to control for confounders, such as matching, stratification and regression modelling, to reduce the risk of bias of confounding variables [52]. In other words, the issues discussed above should be seriously addressed and the methods should be reported more clearly [43].

Few validated measures were used in the studies reviewed, which limited the generalisability of the results. To evaluate MHL, questionnaires were developed for the target population presented above, notably the Mental Health Literacy Scale [53], the Multicomponent Mental Health Literacy Scale [54], and the Mental Health Literacy Questionnaire [55]. Future studies should apply these reliable and valid measures tested with athletes to assess MHL. In addition, help-seeking and stigma measures were used based on self-developed questionnaires, which is recognised as a study limitation for programme evaluation. Therefore, questionnaires with acceptable reliability and validity should be used in future studies to improve their quality.

### 4.6. Consideration for Intervention Delivery Methods

The frequency and duration of the sessions for each programme varied across the studies. For face-to-face interventions, at least one hour with one session was implemented [17]. For the online intervention session lasted at least 10 min [32]. For future MHL programmes with athletes, the frequency and duration of the sessions, including their effectiveness in improving mental-health help-seeking and practical situations, should be taken into consideration, which will make future interventions more effective when tailored to specific target groups.

Three studies were delivered through traditional face-to-face interventions [15,17,31], and two interventions were conducted online [22,32]. For help-seeking behaviour, it was not possible to compare the effectiveness of face-to-face interventions and online interventions, as only one online intervention [22] was evaluated in this review. However, studies have shown that online interventions are not as effective as face-to-face interventions for behaviour change [34,56]. Therefore, future research should further investigate the effectiveness of online interventions to enhance mental health help-seeking.

### 4.7. Limitations and Recommendations

This review has several limitations. The quality of the studies varied, so the results should be interpreted with caution. Many studies applied outcome measures with unclear validity. In addition, some statistical reports were not calculated for unknown information. We are not able to conduct a meta-analysis due to the heterogeneity of operationalisation, measurement, and statistical reporting of the constructs lacked consistency and methodological rigour. Moreover, relevant studies of athletes in other languages may not have been included in this review. Furthermore, we have not used the Medical Subject Headings (MeSH) terms in Medline and equivalent terms in the searched databases. Therefore, some relevant studies may have been missed. Future studies should involve these in search strategies to include more potential studies.

Despite these limitations, the results have practical implications. Interventions with integrative strategies that focus on several barriers to help-seeking should be developed. In addition, psychoeducation designed to enhance MHL should lead to improvements in help-seeking. In addition, reducing the stigma associated with mental disorders or negative attitudes toward help-seeking should be included in programme development [34].

This review also highlighted significant research gaps. First, high-quality RCTs and non-randomised studies with large representative samples and long-term follow-up assessments are needed to provide evidence of the efficacy of interventions. Future research should discuss the active components and cost-effectiveness of mental health help-seeking interventions [34]. Second, most studies did not use reliable and valid measurements to assess the relevant outcomes. Third, only one study discussed mental health behaviour and the programme was supported by the process framework based on the help-seeking theory [9]. In the sport context, the theory of planned behaviour [57] was applied in the MHL intervention for coaches [58]. Moreover, the health belief model [59] was used to explore student-athletes’ mental health help-seeking experience [60]. Therefore, theory-based guidelines for developing and evaluating interventions should be taken into account in future programmes. Finally, all included studies were conducted in Western countries. More research in Eastern countries is needed.

Research on MHL in sport to improve knowledge, help-seeking attitudes, intentions and behaviours and to reduce stigma is still in its early stage. However, it is encouraging to observe the promising results of relevant studies. Future research should take into account the complexity of the sport context when considering appropriate research methodologies for programme evaluation.

## 5. Conclusions

MHL interventions could enhance help-seeking attitudes and intentions, mental health knowledge, stigmas, while not increase help-seeking behaviours. Because MHL programmes are a relatively new and developing research field, researchers and sport practitioners should emphasise that there is still much work to be done in future studies. Although evidence was found in the programmes available, some studies had high risk of bias. Therefore, the methodological quality of studies, including RCTs and other designs, should be improved in future research. Thus, future research on MHL programmes for athletes should focus on theory-based programme development, longitudinal studies, blinded approaches, larger samples of male and female athletes, and validated measurement tools.

## Figures and Tables

**Figure 1 ijerph-17-07263-f001:**
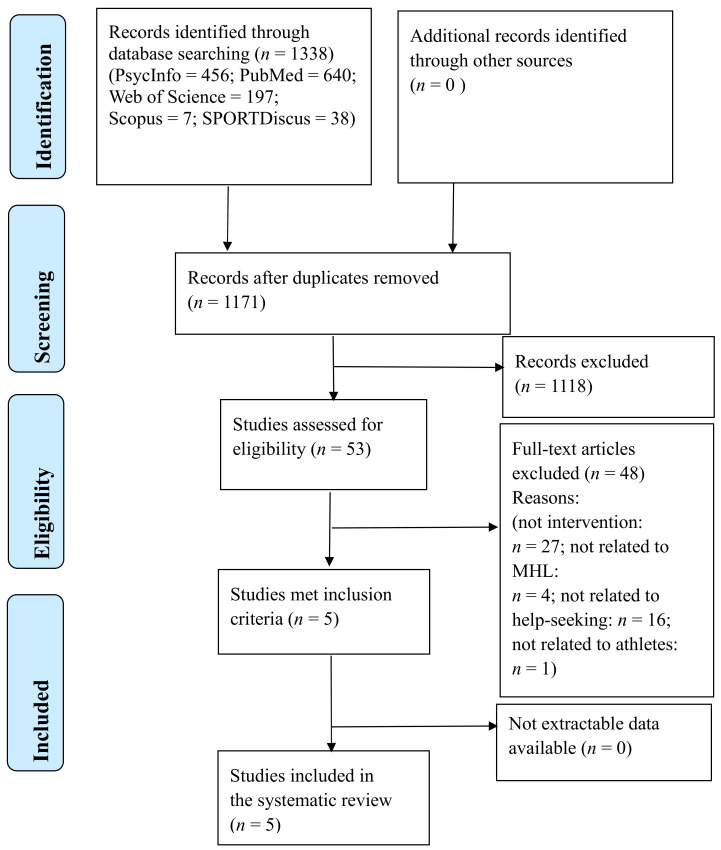
PRISMA 2009 flow diagram. MHL is mental health literacy.

**Table 1 ijerph-17-07263-t001:** Characteristics of included programs and their studies.

First Author (Year)	Country	Athlete Type	Participants (*n*)	% Female	Mean Age (SD), Range	Type of Sport
Breslin et al. [15]	IR	Student-athletes	Pre: 100; Post: 100; Follow-up: 15	41%	20.78 ± 2.91 Range: Not reported	Soccer, Gaelic football, rugby, hockey, netball, and golf
Gulliver et al. [22]	AU	Elite athletes	Pre:59; Post: 59; Follow-up: 40	72.9%	25.5 (SD: Not reported) (median 24.5) Range = 18–48	Not reported
Kern et al. [17]	US	Student-athletes	Pre:652 *; Post: 626	25.1% (unknown: 46.6%)	M (SD): Not reported Ranged = 18–23	Not reported
Pierce et al. [31]	AU	Elite athletes	Pre:275; Post: 96	Not reported	M (SD): Not reported The median age = 21 Range 15–50	Football
Van Raalte et al. [32]	US	Student-athletes	Pre: 153; Post: 146;	Pre: 70.5%; Post: 67.3%	19.63 (SD = 1.76) Range: Not reported	Baseball, fencing, football, basketball, soccer, lacrosse, rugby, skiing, softball, squash, gymnastics, track and field, swimming and diving, tennis, golf, and volleyball.

Country: AU: Australia, IR: Ireland, US: United States of America; Pre: Pre-test; Post: Post-test; M: Mean; SD: standard deviation; *: The Kern et al. Study reported 652 in the baseline sample characteristics, while reported 626 student-athletes completed the pre- and post-surveys in the abstract. To calculate the number of females and males, 652 were used in the pre-test.

**Table 2 ijerph-17-07263-t002:** Methods of included programs and their studies.

First Author (Year)	Intervention Deliverer	Study Design	Control Group	Delivery Method	Follow-Up Points	Duration Length * *n* Sessions
Breslin et al. [15]	Experienced mental health and well-being tutors	CT	Active	Face-to-face	Post, 3-month *	1.25 h * 1
Gulliver et al. [22]	N/A	RCT	Active	Online	Post, 3-month	Available 24 h a day, seven days a week
Kern et al. [17]	Members of the intervention team	Pre- and post- design	No comparison	Face-to-face	Post	1 h * 1
Pierce et al. [31]	A MHFA qualified instructor	CT	unknown	Face-to-face	6-month	12 h
Van Raalte et al. [32]	N/A	RCT	Active	Online	Post	One day (online session lasted at least 10 min)

RCT: Randomized Controlled trial; CT: Controlled trial; N/A: Not applicable; MHFA: Mental Health First Aid; * Only 15 participants finished 3-month follow-up data collection, which not provided in the statistical analyses.

**Table 3 ijerph-17-07263-t003:** Intervention contents of included programs and their studies.

First Author (Year)	Summarize Major Elements
Breslin et al. [15]	(1) Mental health information; (2) seeking help; (3) mindfulness practice; and (4) resilience
Gulliver et al. [22]	(1) Mental health information; (2) seeking help; (3) stigma; and (4) two effective depression treatments
Kern et al. [17]	Mental health information; (2) seeking help; (3) stigma; and (4) supporting friends
Pierce et al. [31]	(1) Mental health information; and (2) seeking help;
Van Raalte et al. [32]	(1) Mental health information; (2) seeking help; (3) stigma; and (4) information about referral

**Table 4 ijerph-17-07263-t004:** Effects of the interventions on mental health knowledge, help-seeking attitudes, intentions, behaviours, and stigmas.

Author (Year)	Effect Size								
	Knowledge		Help-Seeking Attitudes	Help-Seeking Intentions		Help-Seeking Behaviours		Stigma	
Breslin et al. [15]	d ^a^ = 0.16 ***		Not reported	d ^a^ = 0.06 ***		Not reported		Not reported	
Gulliver et al. [22]	d_1_ ^a^ = 0.88 ** d_1_ ^b^ = 0.74 **	d_2_ ^a^ = 0.88 ** d_2_ ^b^ = 0.77 **	d ^a^ = 0.29 d ^b^ = −0.14	d_3_ ^a^ = 0.20 d_3_ ^b^ = 0.05	d_4_ ^a^ = −0.04 d_4_ ^b^ = −0.36	OR_5_ = 57.38, 95% CI_5_ 0.85–3868.09 ^a^ OR_5_ =3.48, 95% CI_5_ 0.10–122.32 ^b^	OR_6_ = 0.74, 95% CI_5_ 0.03–19.12 ^a^ OR_6_ = 0.21, 95% CI_5_ 0.01–7.79 ^b^	d_7_ ^a^ = 0.25 * d_7_ ^b^ = 0.09 **	d_8_ ^a^ = 0.03 ** d_8_ ^b^ = 0.53 *
Kern et al. [17]	N/A		d ^a^ = 0.44 ***	Not reported		Not reported		d ^a^ = 0.20 ***	d ^a^ = 0.06
Pierce et al. [31]	Not reported		N/A	Not reported		Not reported		N/A	
Van Raalte et al. [32]	d^a^ = 0.24 *		Not reported	Not reported		Not reported		Not reported	

* *p* < 0.05, ** *p* < 0.01, *** *p* < 0.001; CI, confidential interval; OR, odds ratio; ^a^ and ^b^ denoted post-test and follow-up test, respectively; N/A: Not applicable; Effect size of the Breslin et al. (2018) study was calculated by the author; In the Gulliver et al. (2012) study, 1 and 2 denoted depression literacy, and anxiety literacy, respectively; 3 and 4 denoted help-seeking intentions from formal sources, and intentions from informal sources, respectively; 5 and 6 denoted help-seeking behaviours from formal help, and behaviours from informal help, respectively; 7 and 8 denoted depression stigma, and anxiety stigma, respectively; The Pierce et al. (2010) study could not calculate the effect size, because the number of participants in each group was not provided; Effect size of the Kern et al. (2017) study is pre-post effect size, and the others are between-group effect sizes.

**Table 5 ijerph-17-07263-t005:** Effects of the interventions on additional outcomes.

Author (Year)	Variable	Intervention Group (Mean (SD))		Control Group [Mean (SD)]		Effect Size
		Pre-Test	Post-Test/Follow-Up Test	Pre-Test	Post-Test/Follow-Up Test	
Breslin et al. [15]	Resilience	3.39 (0.60)	3.37 (0.74) ^a^	3.66 (0.72)	3.62 (0.78) ^a^	d ^a^ = 0.33
	Well-being	26.49 (3.67)	27.32 (4.31) ^a^	26.36 (4.05)	26.81 (4.85) ^a^	d ^a^ = 0.11
Kern et al. [17]	Supporting teammate: Q1	2.79 (1.01)	3.10 (0.95) ^a^	-	-	d ^a^ = 0.30 ***
	Supporting teammate: Q2	2.62 (0.87)	3.00 (0.72) ^a^	-	-	d ^a^ = 0.47 ***
	Supporting teammate: Q3	2.75 (1.23)	3.41 (0.73) ^a^	-	-	d ^a^ = 0.70 ***
Van Raalte et al. [32]	Self-efficacy	8.30 (1.94)	9.03 (1.67) ^a^	8.15 (1.99)	7.69 (2.77) ^a^	d ^a^ = 0.59 **

** *p* < 0.01, *** *p* < 0.001; ^a^ denoted post-test; Q1, Q2, and Q3 meant the three-item of the questions which evaluated the supporting teammate.

**Table 6 ijerph-17-07263-t006:** Risk of bias for randomized studies using the Cochrane risk-of-bias tool.

Study	Criteria 1	Criteria 2	Criteria 3	Criteria 4	Criteria 5	Criteria 6	Criteria 7	Classification
Gulliver et al. [22]	+	+	+	?	+	+	+	+
Van Raalte et al. [32]	?	?	?	?	+	+	+	?

Criteria: (1) Random sequence generation; (2) Allocation concealment; (3) Blinding of participants and personnel (4) Blinding of outcome assessment; (5) Incomplete outcome data; (6) Selective reporting; (7) Other bias; + = Low risk of bias; = High risk of bias; ? = Unclear risk of bias.

**Table 7 ijerph-17-07263-t007:** Risk of bias for non-randomized studies using the Newcastle–Ottawa quality assessment scale.

Study	Selection	Comparability	Exposure (for Study 1 and 3)/Outcome (for Study 2)	Classification
Breslin et al. [15]	****	*	**	Low risk of bias
Kern et al. [17]	*****	-	*	Low risk of bias
Pierce et al. [31]	***	-	-	High risk of bias

* stars identify the level of the quality of the study; Remarks: The more stars * each component gets, the higher quality its represent; A maximum of one ‘star’ for each item within the ‘Selection’ and ‘Exposure/Outcome’ categories; maximum of two ‘stars’ for ‘Comparability’.

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
