# Peer review of "Mental Health Literacy Intervention on Help-Seeking in Athletes: A Systematic Review"

_ijerph, 2020, doi:10.3390/ijerph17197263_

Round 1
Reviewer 1 Report
This manuscript reports on a systematic review of studies of the effects of mental health literacy (MHL) interventions for athletes. The topic addressed in the manuscript is timely and falls within the domain covered by this journal. The methods undertaken by the authors are thorough and comprehensive, and the conclusions are reasonable based on the findings of the systematic review. These positive impressions notwithstanding, I have several concerns about the manuscript in its current form:
1. The manuscript could benefit from a thorough proofreading and editing by a native English-speaker. Minor changes can be made to greatly enhance the readability of the manuscript. For example, “was” should be “is” on l. 13, “the” should be deleted on l. 15 and 17, it should be “programs” on l. 15, it should be “a null effect” on l. 22, and it should be “for” instead of “in” on l. 22. Many such changes should be made throughout the manuscript.
- Regarding the material on l. 25, although MHL interventions have not yet been shown to increase help-seeking behaviors, they certainly could be found effective in the future.
- The findings from the systematic reviews performed in references 22 and 23 should be described on l. 65 and 68, respectively.
- With respect to the keywords identified on l. 85-91, the second set of keywords (i.e., “sport organization OR sport club…”) seems overly restrictive. How many studies would there have been without the second set of keywords?
- Shouldn’t the number “53” on l. 154 be “80” (i.e., 1,171-1,091)?
- A word seems to be missing in Figure 1 after “Studies assessed for.”
- The participant categories listed on l. 164-165 are not mutually exclusive (e.g., club athletes can also be elite athletes).
- In Table 1, it should be “Not reported” in multiple locations.
- In many locations throughout the manuscript (especially from l. 255-282), the word “the” should precede the author of a particular study (e.g., “the Pierce et al. study [30]” on l. 255).
Author Response
Review 1
This manuscript reports on a systematic review of studies of the effects of mental health literacy (MHL) interventions for athletes. The topic addressed in the manuscript is timely and falls within the domain covered by this journal. The methods undertaken by the authors are thorough and comprehensive, and the conclusions are reasonable based on the findings of the systematic review. These positive impressions notwithstanding, I have several concerns about the manuscript in its current form:
- The manuscript could benefit from a thorough proofreading and editing by a native English-speaker. Minor changes can be made to greatly enhance the readability of the manuscript. For example, “was” should be “is” on l. 13, “the” should be deleted on l. 15 and 17, it should be “programs” on l. 15, it should be “a null effect” on l. 22, and it should be “for” instead of “in” on l. 22. Many such changes should be made throughout the manuscript.
Response:
Thank you for the comment. We have revised it accordingly (please refer to the revised manuscript in the following pages: Page 1 Line 13, 15, 24, 25).
- Regarding the material on l. 25, although MHL interventions have not yet been shown to increase help-seeking behaviors, they certainly could be found effective in the future.
Response:
Thank you for the comment.We have revised it “MHL interventions can enhance help-seeking attitudes and intentions and mental health knowledge and reduce stigma, but do not increase help-seeking behaviours for now. Further studies should evaluate interventions to enhance help-seeking behaviours” (please refer to the revised manuscript in the following pages: Page 1 Line 28-29).
- The findings from the systematic reviews performed in references 22 and 23 should be described on l. 65 and 68, respectively.
Response:
Thank you for the comment. The findings of the reference 22 you mentioned, which is reference 23 now, were added. However, after serious consideration, we deleted original reference 23 from our revised manuscript, because this review was not focused on sport context. Therefore, only one findings were described. Thank you so much for your understanding (please refer to the revised manuscript in the following pages: Page 2 Line 71-74).
- With respect to the keywords identified on l. 85-91, the second set of keywords (i.e., “sport organization OR sport club…”) seems overly restrictive. How many studies would there have been without the second set of keywords?
Response:
Thank you for the comment. We researched the 5 database without the second set of keywords (i.e., “sport organization OR sport club…”) as you suggested, the search data due to August 31, 2020. Records identified through database searching (n = 5494) (PsycInfo =2100; PubMed = 3049; Web of Science = 286; Scopus = 11; SPORTDiscus = 48). We tried to specific all the articles in sports, therefore, this set of keywords were searched. Furthermore, we searched from the references in the included articles and other potential sources, no further eligible studies included. Thank you for your valuables suggestions again.
- Shouldn’t the number “53” on l. 154 be “80” (i.e., 1,171-1,091)?
Response:
Thank you for the comment. Actually, after screening the full text of the remaining 53 studies, only five studies met the inclusion criteria. Therefore, the number is 53. (please refer to the revised manuscript in the following pages: Page 4, Line 156-157).
- A word seems to be missing in Figure 1 after “Studies assessed for.”
Response:
Thank you for the comment. We have reinserted Figure. 1 (please refer to the revised manuscript in the following pages:Page 5 Line 163).
- The participant categories listed on l. 164-165 are not mutually exclusive (e.g., club athletes can also be elite athletes).
Response:
Thank you for the comment. We have revised it to “The participants were elite athletes (N = 2), and student-athletes (N = 3).” (please refer to the revised manuscript in the following pages: Page 5 Line 170-171, Page 7 Line 191).
- In Table 1, it should be “Not reported” in multiple locations.
Response:
Thank you for the comment. We have revised it based on your valuable suggestions (please refer to the revised manuscript in the following pages: Page 7 Line 191).
- In many locations throughout the manuscript (especially from l. 255-282), the word “the” should precede the author of a particular study (e.g., “the Pierce et al. study [30]” on l. 255).
Response:
Thank you for the comment. We have revised it accordingly (please refer to the revised manuscript in the following pages: Page 5 Line 181-182, 188; Page 6 Line 189; Page 9 Line 211, 213, 223, 226-227, 230-231, 237; Page 12 Line 247, 254-255, 257, 263-264, 266-267, 273-275, 279; Page 14 Line 300, 324, 331-332, 334-335; Page 15 Line 337, 342, 369, 375, 377; Page 16 Line 385).

Reviewer 2 Report
The paper is about literacy, which is the recognition of mental health problems, in a very particular category of workers, athletes. These individuals are particularly sensitive to the stigma associated with mental health problems.
The authors carry out a systematic review of the literature applying in an exemplary way the principles of this type of investigation.
I have no advice to give because the procedure has been correctly followed. My native language is not English and therefore I cannot recommend language improvements.
The main weakness of the study is the low number of selected articles and their low quality. This defect is not the fault of the authors, it cannot be eliminated. The authors have correctly indicated this point in the limitations. The strength of the work is in its originality, as well as in the methodological scruple.
Unfortunately, only 5 jobs are available and not all of them are of high quality. This limits the value of the evidence collected. The results are discussed in depth.
Author Response
Reviewer 2
The paper is about literacy, which is the recognition of mental health problems, in a very particular category of workers, athletes. These individuals are particularly sensitive to the stigma associated with mental health problems.
The authors carry out a systematic review of the literature applying in an exemplary way the principles of this type of investigation.
I have no advice to give because the procedure has been correctly followed. My native language is not English and therefore I cannot recommend language improvements.
The main weakness of the study is the low number of selected articles and their low quality. This defect is not the fault of the authors, it cannot be eliminated. The authors have correctly indicated this point in the limitations. The strength of the work is in its originality, as well as in the methodological scruple.
Unfortunately, only 5 jobs are available and not all of them are of high quality. This limits the value of the evidence collected. The results are discussed in depth.
Response:
Thank you for the comments. We really appreciate the positive feedback and encouragement from you. We will try our best to work better in this area.

Reviewer 3 Report
Thank you for the opportunity to review this interesting paper. I have some comments which I believe could further strengthen the paper-
- Abstract- The abstract does not contain enough methodological background.
- Introduction- The introduction mainly discusses about mental help seeking behaviour but it does not explore the underlying factors causes mental health illnesses among the athletes. The rationale for doing the study is also very weak. If there is limited knowledge on MHL intervention, then how this review was conducted? Hence this is completely inaccurate.
- Methodology- Page 2 line 90 the filter option in the sidebar of each database is used. You needed to mention those options. Page 5 Figure 1, please include the number for studies assessed and no extractable data
- Analysis- The analysis process is not clearly stated. It seems like you only done summary of the findings with some descriptive analysis and it does not reveal anything.
- Findings- Not congruent with the research objective.
Author Response
Reviewer 3
Thank you for the opportunity to review this interesting paper. I have some comments which I believe could further strengthen the paper-
- Abstract- The abstract does not contain enough methodological background.
Response:
Thank you for the comments. Methodological information was added, such as five electronic databases were systematically searched for articles published before May 2020. The selection procedure was based on the PRISMA guidelines.All kinds of study designs were included. Risk of bias was assessed for each study using the Cochrane tool and the Newcastle–Ottawa quality assessment scale (please refer to the revised manuscript in the following pages: Page 1 Line 17-19, 21-22).
- Introduction- The introduction mainly discusses about mental help seeking behaviour but it does not explore the underlying factors causes mental health illnesses among the athletes. The rationale for doing the study is also very weak. If there is limited knowledge on MHL intervention, then how this review was conducted? Hence this is completely inaccurate.
Response:
Thank you for the comments. Firstly, in the introduction part, the purpose of this study is to explore help-seeking behaviours, therefore the underlying factors causes mental health illnesses among the athletes were not further discussed. But we added the information about the severe situation of mental health issues in athlete (please refer to the revised manuscript in the following pages: Page 1 Line 38-42).
Secondly, we revised the manuscript and further justify the rationale of this study. Due to the IOC’s increasing awareness of the importance of addressing mental health issues (Reardon et al., 2019),several programmes have been developed to address these barriers and promote help-seeking. As such, a systematic review of the existing evidence is required to evaluate the efficacy of MHL interventions. In response to increasing MHL interventions in the sport context and to improve the methodological transparency of previous MHL intervention studies, we conducted a systematic review of MHL programmes that focused on athletes and aimed to enhance mental health knowledge, improve help-seeking attitudes, intentions and behaviours and reduce stigma (please refer to the revised manuscript in the following pages: Page 2 Line 67-70, 75-78).
- Methodology- Page 2 line 90 the filter option in the sidebar of each database is used. You needed to mention those options. Page 5 Figure 1, please include the number for studies assessed and no extractable data
Response:
Thank you for the comments. Considering included more studies, the filter option in the sidebar of each database was limits in English. No further limit used to screening articles. In order to avoid the confusion, we have deleted this sentence and only stated that “In the current review, we focused on articles published in English” in the Eligibility Criteria part (please refer to the revised manuscript in the following pages: Page 3 Line 96). Furthermore, we reinserted the figure 1 (please refer to the revised manuscript in the following pages: Page 5 Line 163).
- Analysis- The analysis process is not clearly stated. It seems like you only done summary of the findings with some descriptive analysis and it does not reveal anything.
Response:
Thank you for the comments. Since this article is a systematic review, no additional analyses were conducted, including separate meta-analyses or subgroup and sensitivity analyses (please refer to the revised manuscript in the following pages: Page 4 Line 151-152). Furthermore, the whole data analysis part was followed the PRISMA guidelines, which could help this article presented clearly. Future studies might conduct the meta-analysis if more MHL interventions focused on athletes published, which also stated in the limitation of this study (please refer to the revised manuscript in the following pages: Page 17 Line 433-435).
- Findings- Not congruent with the research objective.
Response:
Thank you for the comments. In response to increasing MHL interventions in the sport context and to improve the methodological transparency of previous MHL intervention studies, we conducted a systematic review of MHL programmes that focused on athletes and aimed to enhance mental health knowledge, improve help-seeking attitudes, intentions and behaviours and reduce stigma (please refer to the revised manuscript in the following pages: Page 2 Line 75-78). The findings demonstrated that MHL interventions could enhance help-seeking attitudes and intentions, mental health knowledge, stigmas, while not increase help-seeking behaviours (please refer to the revised manuscript in the following pages: Page 17 Line 458-459).

Round 2
Reviewer 1 Report
I appreciate the efforts of the authors in addressing my concerns. The manuscript has been strengthened by the changes that have been made. My sole remaining concerns are minor:
1. On line 44, it should be "did not seek professional assistance."
2. In Tables 1 and 4, it should be changed from "Not report" to "Not reported" in multiple locations.
3. I have no reason to believe that 53 articles were not examined for eligibility, but I am still not clear why there were 53 articles remaining after 1,091 articles were removed from a total of 1,171 articles. Basic mathematics suggests that there were 80 articles remaining. Please explain what happened to the "missing" 27 articles.
Author Response
Reviewer 1:
- On line 44, it should be "did not seek professional assistance."
Response:
Corrected (please refer to the revised manuscript in the following pages: Page 2 Line 44).
- In Tables 1 and 4, it should be changed from "Not report" to "Not reported" inmultiple locations.
Response:
Corrected (please refer to the revised manuscript in the following pages: Page 7 Line 192, Page 10 Line 241).
- I have no reason to believe that 53 articles were not examined for eligibility, but I am still not clear why there were 53 articles remaining after 1,091 articles were removed from a total of 1,171 articles. Basic mathematics suggests that there were 80 articles remaining. Please explain what happened to the "missing" 27 articles.
Response:
Thank you for the comments. We have carefully checked the original data in Endnote. We found that there was a mistake when calculating the number. We have revised the No. From 1091 to 1118 (please refer to the revised manuscript in the following pages: Page 4 Line 157, Page 5 Line 164).
Reviewer 3 Report
The paper is methodologically flawed and I don't think it suitable for publication in your journal so I rejected it when I first reviewed it.
Author Response
Thank you for the comments. We will try our best to work better in this area.